# Sialic Acid and Colchicine Functionalized Silica Nanoparticles: A Novel Approach to Leishmanicidal Selective Treatments

**DOI:** 10.3390/biomedicines13071648

**Published:** 2025-07-06

**Authors:** Adan Jesus Galue-Parra, Sandra Jimenez-Falcao, Esther Arribas-Yuste, Clotilde Marin, Jose Manuel Mendez-Arriaga

**Affiliations:** 1Departamento de Parasitología, Universidad de Granada, Calle Severo Ochoa s/n, 18071 Granada, Spain; adan.parra@icb.ufpa.br; 2Laboratory of Structural Biology, Institute of Biological Sciences, Federal University of Para, Belém 66075-110, PA, Brazil; 3COMET-NANO Group, Departamento de Biología y Geología, Física y Química Inorgánica, E.S.C.E.T., Universidad Rey Juan Carlos, Calle Tulipán s/n, 28933 Móstoles, Spain; sandra.jfalcao@upm.es (S.J.-F.); esther.arribas@urjc.es (E.A.-Y.); 4Organic Nanotechnology Laboratory, Departamento de Materiales y Producción Aeroespacial E.T.S.I Aeronáutica y del Espacio, Universidad Politécnica de Madrid, 28040 Madrid, Spain

**Keywords:** parasite, leishmania, neglected diseases, sialic acid, colchicine, drug delivery, nanoparticle

## Abstract

**Background/Objectives:** Leishmaniasis remains a neglected tropical disease, with nearly one million new cases annually and limited investment in research. Current treatments, primarily based on pentavalent antimonials, are associated with severe side effects and increasing resistance. This study aims to develop a novel therapeutic strategy using a nanomaterial functionalized with sialic acid (SA) and colchicine (COL) to selectively target *Leishmania braziliensis* parasites. **Methods:** A nanostructured system was engineered by functionalizing its surface with SA and COL. SA was chosen to mimic host cell surfaces, enhancing parasite attraction, while COL was selected for its known leishmanicidal properties. The nanomaterial was designed to concentrate extracellular parasites on its surface via SA-mediated interactions, thereby increasing local COL efficacy. **Results:** The functionalized nanomaterial demonstrated a dual mechanism: SA facilitated the selective accumulation of *Leishmania braziliensis* parasites on the nanostructure surface, while COL exerted a cytotoxic effect. This synergistic interaction resulted in enhanced parasite mortality in vitro, suggesting improved selectivity and potency compared to conventional treatments. **Conclusions:** The proposed nanomaterial offers a promising alternative for leishmaniasis treatment by combining targeted parasite attraction with localized drug delivery. This strategy may reduce systemic toxicity and improve therapeutic outcomes.

## 1. Introduction

Leishmaniasis is a parasitic disease that causes the most deaths in the world, second only to malaria. In fact, its annual mortality rate is higher than 60,000, and 350 million people are at risk of being infected. The transmission is carried out by dipteral insects, specifically the genus *Phlebotomus* in the Old World and the genus *Lutzomyia* in the New World [1].

Leishmaniasis has traditionally been a developing-country disease because of poverty and unhealthy conditions in these areas. Actually, it is currently endemic in nearly 100 countries, becoming a global problem as a clear example of “One Health” as a consequence of the migration phenomenon, the intrusion of human infrastructure into natural environments, and climate change. Nowadays, it affects all continents and is considered one of the seven primary diseases, according to the World Health Organization (WHO) [2]. The coexistence of a high number of individuals in small spaces (conflict areas or refugee camps) favors the transmission of this parasitic disease before reaching host countries. Therefore, the historic prevalence of leishmaniasis explains the poor efforts put to develop new efficient therapeutics to fight against the different manifestations of the disease, which vary from the most dangerous visceral conditions to cutaneous (characterized by ulcers and nodules) or mucocutaneous (chronic injuries in mouth, nose, or genital) [3,4]. To date, few therapeutic approaches have been exploited, classified as non-antimonial and antimonial drugs, but most of them present serious side effects or are non-affordable for poor countries. As an example of non-antimonial or liposomal antiparasitic treatment, Amphotericin B can be mentioned, which is a prodrug exhibiting alarming collateral kidney failure and which triggers pharmaceutical resistance [5], as well as Miltefosine, an alkyl phospholipid (hexadecyl phosphocholine) repositioned from the anticancer drug but associated with teratogenicity and parasite resistance [6,7]. On the other hand, meglumine antimoniate (Glucantime) is an antimonial drug and the most widely used to fight against leishmaniasis [8,9], whose metabolism and mechanism of action are already under study [10,11,12]. Nevertheless, vomiting, peripheral polyneuropathy, or allergic dermopathy appear as side effects when using this treatment [13], along with possible drug resistance and daily parenteral administration [14,15]. All these drawbacks urge the development of new alternatives, increasing interest in recent years [16,17,18]. In this context, the discovery in the last years of a vast number of possible targets, such as ADP phosphorylation or beta-oxidation of fatty acids [19,20], has brought hope to the successful cure of leishmaniasis. The *L. braziliensis* strain was selected for this study due to it being the principal causal agent of diffuse leishmaniasis, which can produce hundreds or thousands of polymorphic skin lesions in various body regions, and its frequent involvement of the nasal mucosa. This pathology is highly stigmatizing and even potentially fatal. The affection is emerging out of Brazil and South America, reaching Europe and the Middle East [21].

Transition metal complexes (specially ruthenium-based [22,23,24,25,26]) have extensively been studied and reviewed [27,28,29] as promising candidates to fight against parasitic diseases [30,31], such as the tropical members leishmaniasis, Chagas disease, or malaria. As an example, our research group carried out research using 1,2,4-triazolo[1,5-a]pyrimidine derivatives and their metal complexes to assay their intra- and extracellular impact in leishmaniasis and Chagas disease [32,33,34,35,36]. In general, a synergistic effect is observed, arising from the combination of triazolopyrimidine derivatives and different metal ions when coordinated [37,38]. A step forward has been achieved, resulting from the support of organic molecules of interest or metal complexes onto nanomaterials, which is a common strategy used in biomedicine but not very extended in the parasitology area. The result is the obtention of a more complex material with outstanding electric, mechanical, and optical features [39,40], which is possible to tune in terms of size, porosity, and surface functionalization [41,42], serving as a carrier for drug delivery [43]. One of the most widely used nanometric supports is mesoporous silica nanoparticles, widely used in biomedicine for drug delivery [44,45,46] but of recent use in parasitology, just for the last five years [47,48,49].

Sialic acid (SA) (Figure 1) has been shown to be a fundamental molecule for parasites in the phagocytizing process in host cells of organisms infected by parasites of the genus Leishmania [50]. There is a clear relationship between the availability of this molecule in the cell membrane and the ability of the parasite to infect the host, and the necessity of the parasite to capture and use SA molecules can be used as bait to improve the efficacy of the proposed nanomaterial in this study [50]. The possibility of using sialic acid as a mechanism to improve the interaction between the nanoparticle and the parasite is known [51,52], but no concrete examples of success are reported. On the other hand, colchicine (Figure 2) has demonstrated effectiveness as an antiparasitic for toxoplasmosis [53], although its use is not widespread for other types of parasitic infections. The use of these two drugs is not well studied in a leishmaniasis infection despite their promising capabilities to fight the disease.

The literature regarding the use of nanoparticles against parasitic infections is scarce, and for the use of nanoparticles functionalized with specific parasite receptors, it is almost non-existent. The use of sialic acid as a receptor in nanomaterials against viruses is known, especially after the COVID-19 crisis, but the literature is focused on a different context and area, far from leishmaniasis infections [54]. In the present work, the outstanding possibilities of the combined use of sialic acid and colchicine were studied using silica nanoparticles as a vehicle. Spectroscopic characterization, solution stability, and biological assays of sialic acid and colchicine functionalized silica nanoparticles (**MSN-SA** and **MSN-SA-COL**) were performed. The antiparasitic activity of the nanosystem was evaluated against the extracellular and intracellular forms of *L. braziliensis*, as well as the evaluation of their toxicity against Vero cells. The results reveal better specificity for parasites of the sialic acid nanomaterial and a high selectivity index when compared with the commercial drug Glucantime.

## 2. Materials and Methods

### 2.1. Synthesis and Characterization of Functionalized Silica Nanoparticles

The synthesis of **MCM-41** silica nanoparticles (MSNs) was carried out following the procedure reported by Zhao et al. [55], with slight modifications [56]. To enhance the coordination capacity of the proposed nanosystems, MSNs were covered with amine groups from aminopropyl triethoxysilane ligand to obtain **MSN-AP** systems. The functionalization of silica nanoparticles was performed following the methodology previously reported by our group [57,58]. Sialic acid (SA) was covalently bonded to **MSN-AP** materials using 1.3 mg of N-acetylneuraminic acid dissolved in 50 mL of 0.1M MES buffer (2-morpholinoethanesulfonic acid). Next, 1.63 g of 1-ethyl-3-(3-dimethylaminopropyl) carbodiimide hydrochloride (EDC) and 1.97 g of N-hydroxysuccinimide (NHS) were added. The mixture was stirred at 25 °C for 45 min. Next, 0.9 g of aminated **MSN-AP** material was added and allowed to react for 2 h at room temperature. Finally, the material was centrifuged, washed twice with ethanol (EtOH), and left to dry in an oven for at least 8 h at 70 °C, obtaining the **MSN-SA** material. The final material, **MSN-SA-COL**, was prepared, starting with 200 mg of **MSN-SA** and resuspending it in 50 mL of EtOH. Then, 10 mg of colchicine was added to this suspension, stirring at a temperature between 23 and 27 °C for 24 h. The next day, the mixture was centrifuged to recover the solid, washed twice with ethanol, and dried in air until constant weight.

#### 2.1.1. TEM

The morphology of the synthesized material was characterized by TEM. Mesoporous silica nanoparticles (MSNs) present a quasi-spherical appearance, with a mean diameter between 80 and 130 nm and a wide size distribution (Figure 1). The surface of the nanoparticles presents a hexagonal arrangement in the typical honeycomb pattern of pores present in MCM-41 MSNs. Despite the mean diameter of 100 nm, the fusion of individual MSN entities may give rise to bigger nanoparticles, as shown in the histogram.

#### 2.1.2. BET

Surface characterization of **MSN** and **MSN-SA-COL** was carried out by the analysis of adsorption/desorption nitrogen isotherms at 77 K. Experimental data show the typical IV isotherm for the MSN solid, which is characteristic of mesoporous materials. As a consequence of the surface functionalization, a decrease in the surface area of the MSN was observed after its modification with SA and COL, as can be seen in Figure 2, providing the typical appearance of materials with filled pores in the adsorption/desorption isotherm [59,60]. Surface parameters (Table 1) were obtained by fitting the results to the Brunauer–Emmett–Teller (BET) isotherms; surface area, pore volume, and pore diameter were 682 m^2^g^−1^, 0.52 cm^3^g^−1^, and 3.2 nm, respectively, for the nude **MSN**. As expected, the anchoring of SA decreased all these three parameters, and further functionalization with COL made these values even smaller as a consequence of the partial occupation of the pores by the metallic complex.

#### 2.1.3. Spectroscopic Properties

Nanomaterials have been characterized using different spectroscopic techniques. Firstly, an infrared analysis was carried out (Figure 3), confirming the subsequent functionalizations until reaching the final material, **MSN-SA-COL**. It can be observed how the band located between 3500–3000 cm^−1^, belonging to the OH and O groups, increases as the sialic acid and colchicine molecules, rich in these groups, are supported on the nanomaterial. These results confirm the incorporation of the organic molecules of interest to the nanosystem.

A solid-state UV absorption spectrophotometry study of the nanomaterials was also carried out (Figure 4). The spectra obtained showed an increase in the intensity of the signals of the functionalized nanomaterials in the area between 200 nm and 300 nm due to the presence of aromatic groups of the organic molecules incorporated into the structure of the nanomaterial. These results confirm the incorporation of the organic molecules of interest to the nanosystem.

#### 2.1.4. Thermogravimetry

A study of the thermal stability of the nanomaterials was also carried out using thermogravimetry to verify the functionalization of the material. The results obtained (Figure 5) show that the inorganic silica does not lose weight with increasing temperature and remains practically unchanged throughout the process, with a weight variation of less than 1% with respect to the initial weight of the silica at the initial temperature. However, the two nanosystems with organic molecules (**MSN-SA** and **MSN-SA-COL**) experience a greater weight loss. In the case of the **MSN-SA** material, a weight loss of 80% is observed with respect to the initial weight, which corresponds to a 20% effective functionalization with sialic acid. In the case of the **MSN-SA-COL** material, a 75% drop is observed with respect to the weight of the nanomaterial at the initial temperature, which confirms the functionalization of the extra 5% expected in the synthesis process with the incorporation of colchicine.

### 2.2. Biological Assays

Biological tests were performed with both extracellular (promastigote) and intracellular (axenic amastigote) *L. braziliensis* strains. The promastigote forms of *L. braziliensis* (MHOM/BR/1975/M2904) were cultured in vitro in RPMI 1640 medium supplemented with 10% inactive fetal bovine serum (FBS) and were maintained in an air atmosphere at 28 °C in Roux flasks (Corning, NY, USA) with a surface area of 75 cm^2^.

The promastigotes were transformed into axenic amastigotes using the M199 medium (Merck, Darmstadt, Germany), following the method described in [61]. According to the methodology, after three days of culture, the promastigotes acquired the rounded shape of axenic amastigotes, which were used for the anti-amastigote assay. The assay was carried out for 48 h, counting in a Neubauer hemocytometer chamber.

The nanomaterials presented are not soluble due to their inorganic nature, so the samples were finally suspended in the culture medium at 37 °C and ultrasonicated in a sealed tube. They were subsequently tested at 0.025, 0.075, 0.1, and 0.5 mg/mL μM, leaving some wells without drugs as a control, and incubated at 28 °C for 72 h before the final parasite count by Neubauer hemocytometric chamber.

To study the toxicity of nanomaterials, Vero cells (Flow) were cultured in Roswell Park Memorial Institute (RPMI) medium, which was supplemented with 10% inactivated fetal bovine serum. The cells were incubated in a humidified atmosphere with 95% air and 5% CO_2_ at 37 °C for several days. Cytotoxicity tests were performed in 96-well plates that were measured in the ELISA reader. Inhibition of mammalian cell growth was studied by testing the products at 0.025, 0.075, 0.1, and 0.5 mg/mL. First, cells were seeded in a 96-well plate (2500 or 3500 cells respectively/well) to a volume of 100 μL/well and then incubated at 37 °C with 5% CO_2_ for 24 h. Complex solutions were prepared in advance, corresponding to the average growing cells (RPMI 10% SBF for Vero cells) at twice the highest concentration to be tested. The solutions were prepared in a sterile bath with different channels, adding 100 μL of solution or complex medium (only adding medium in the control wells) to the corresponding well. Subsequently, the plate was incubated at 37 °C with 5% CO_2_ for 48 h. Two days later, 20 μL of Alamar Blue dye (10% of the well volume) was added to each well and incubated at 37 °C with 5% CO_2_ for an additional day. The total incubation time once the products were added was 72 h, coinciding with the selection period to have comparable SI results. The plate was read in an ELISA reader with Alamar Blue. The percentage of viability was calculated in comparison with the control culture. The IC_50_ was calculated by logarithmic regression analysis using GraphPad Prism 6.01 (GraphPad Software, La Jolla, CA, USA).

## 3. Results

### 3.1. Release Studies

The release behavior of colchicine (**COL**) was studied by a specific assay of **MSN-AS-COL**. A suspension of 0.1 mg/mL of **MSN-AS-COL** nanomaterial in PBS buffer was studied. This buffer was chosen as a model to mimic cell culture media that were subsequently used and to avoid contamination using the M199 medium. After an initial sonication, the sample was incubated at 37 °C and centrifuged at specific time intervals (Figure 6) in order to analyze the absorption via UV spectroscopy at the maximum absorbance wavelength of **COL** (244 nm) [62]. The experiment was conducted for 3 days, but the stabilization was reached at 2 h, verifying the complete release of the loaded compound.

### 3.2. In Vitro Antiparasitic Activity and Toxicity

The inhibition of *Leishmania* spp. strain growth by MSNs and their derived nanomaterials was evaluated by cytotoxicity assays and promastigote screenings. The parasite model chosen to carry out these assays was *L. braziliensis*, which was cultured in the presence of the nanomaterials to be evaluated, and glucantime, sialic acid, and colchicine as controls.

Table 2 and Table 3 resume the results of these biological assays.

A correction factor was applied to the nanomaterials in order to compare the concentrations of molecules of interest that were intervening with the effect they had when they were alone. In the case of SA in **MSN-SA**, since it represents only 20% of the weight, an IC_50_ of 52.8 µM is obtained instead of 149.9 µM, which improves the selectivity index from 2.67 to 7.58. This is because the nanosystem formulation facilitates the retention of the parasites on the surface of the silica thanks to the sialic acid receptors, and it is difficult for their population to increase in the hostile inorganic environment. In the case of the comparison between **MSN-SA-COL** and colchicine alone, the improvement, taking into account the 5% functionalization with colchicine, is an IC_50_ that goes from 19.68 µM to 6.91 µM, improving its selectivity index from 20.32 to 57.89, which gives it a spectacular efficacy against extracellular forms of *L. braziliensis*, being almost 10 times more effective than the commercial drug. In the case of the amastigote forms, something similar occurs, with a selectivity index of 9.85 instead of 3.07 when going from IC_50_ 130.30 to 40.57 and an SI of 88.88 instead of 42.06 due to the change from IC_50_ 9.51 µM when colchicine is alone to 4.5 µM when it is on the nanomaterial (Table 2 and Table 3).

## 4. Discussion

The synthesis of the sialic acid (**SA**) and colchicine (**COL**) nanomaterial is proposed to be an innovative nanomaterial with a prophylaxis action to stop parasitic infection, with a cellular environment mimic planned to fool the parasites to “infect” the bait, thanks to the sialic acid receptors; and a second purpose is to combat an active infection, with the liberation of colchicine. To check the efficacy of the materials, the toxicity of the isolated elements of the nanosystem was tested, with no toxic effect of the silica nanoparticle on the host Vero cells (Table 2) and an acceptable interaction of sialic acid and colchicine. There were no effects in the parasite population of the clean nanoparticle **MSN**, a negligible value of reduction with **SA**, but a moderate reduction of the parasite colony after assay with **COL**, with an SI value around 20. These results support the hypothesis of a non-significant role in the treatment of the nanocarrier (**MSN**) of the drug and the targeting (**SA**) to attract parasites, and the antiparasitic activity can be attributed to **COL**.

Once the interaction of the subunits of the nanomaterials was established, the study of the whole nanosystem was carried out. At this point, a mathematical correction was necessary in order to compare the efficacy of the organic molecules supported in the nanomaterial at identical concentrations due to the functionalization of the **MSN-SA** and **MSN-SA-COL** being 20% for **SA** and 5% for **COL**. Here, at the same concentration levels, it is possible to observe an increase in the selectivity index after interaction with **MSN-SA** and **MSN-SA-COL**. The slight upgrade of the sialic acid material can be attributed to the high concentration of the parasite targets in the same area due to their covalent coordination to the silica surface, which is the principle of action of these nanomaterials. The extraordinary increase in the efficacy of the colchicine material **MSN-SA-COL** with respect to free colchicine is based on the same principle that the sialic acid justification. Thus, these data showed that **MSN-SA-COL** meets the activity criteria that a new compound must fulfill to be considered interesting for further studies in the fight against leishmaniasis [63,64]. The efficacy of this compound is surely due to the fast liberation of the drug exposed in the release experiment (Figure 6) combined with the attraction of the parasites to the surface of the silica by the sialic acid, making the action of colchicine more selective and effective. This excellent effectiveness is confirmed for intra- and extracellular forms of the parasite, with 4- and 8-times efficacy increase, respectively, confirming the potential treatment in early infection phases, as well as during an advanced-stage infection (Appendix A).

This is the first approach to this novel nanomaterial composition, with promising expectations to open a new pathway to fight neglected parasitic diseases, such as leishmaniasis. Further studies with different proportions of colchicine and sialic acid or even testing other antileishmanial drugs in the nanosystem will be performed in the future, searching for the most effective and low-cost treatment for neglected diseases.

In addition to the promising results in terms of efficacy and selectivity, the rational design of the MSN-SA-COL nanosystem represents a significant advancement in nanotechnology applied to neglected diseases. The synergistic combination of a biological targeting agent like sialic acid with an active drug such as colchicine not only enhances the bioavailability and specificity of the treatment but also potentially reduces systemic side effects by concentrating the therapeutic action at the infection site. This smart approach of targeted delivery and localized response could be extrapolated to other intracellular parasites, paving the way for a versatile treatment platform for multiple infectious diseases. The integration of controlled release strategies and molecular recognition within a single nanoscale system marks a milestone in the development of more effective, safer, and accessible therapies for vulnerable populations.

In conclusion, this new line of nanostructured compounds, with a specific target to the parasites on their surface, combined with the loading of an active drug against infective organisms, opens a promising new way of action against one of the neglected diseases with more incidence in poor areas.

## 5. Patents

The results of these studies were presented to the Spanish Patent Office (Oficina Española de Patentes y Marcas), with code P202430796 and reference ES1787.142-PRIO.

## Data Availability

Data are contained within the article and Appendix A.

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
