# Peer review of "Sialic Acid and Colchicine Functionalized Silica Nanoparticles: A Novel Approach to Leishmanicidal Selective Treatments"

_biomedicines, 2025, doi:10.3390/biomedicines13071648_

Round 1

Reviewer 1 Report

Comments and Suggestions for Authors

This manuscript proposes a targeted treatment strategy based on nanomaterials for leishmaniasis, a neglected tropical disease, which meets the current "One Health" requirements. The technical combination is novel: for the first time, sialic acid (SA, a host cell mimic) and colchicine (COL, a cytotoxic drug) are jointly used in a nanocarrier, enhancing selectivity through a dual mechanism. However, there are some shortcomings.

1.1.Lack of synergistic effect evaluation: No comparison was made between the activities of MSN-SA-COL and free SA+COL, making it impossible to confirm the synergistic advantages of the nanocarrier. No relevant statistical analysis or graphs were provided for this comparison.

2.Incomplete testing scope:Only promastigotes (in vitro) were tested, lacking validation of intracellular amastigotes which better represent the clinical infection stage.

3.Unit inconsistency:IC50 values in the table are presented in both mg/mL and μM, compromising data comparability (e.g., when comparing Glucantime).

4.Incomplete technical details:(1)Reproducibility of synthesis: The consistency of particle size between nanoparticle batches is not described (TEM shows a wide distribution of 80-230 nm).(2)Low loading efficiency: COL loading is only 5%, without discussing optimization potential (e.g., pH adjustment or surface modification).

5.Subheading 2.1.1, 2.1.2: Inconsistent Writing Style.

6.Write the references in accordance with the format of this journal.

7.Strengthen the discussion section.

8.Present the content in accordance with the format specifications of this journal.

Comments on the Quality of English Language

/

Author Response

This manuscript proposes a targeted treatment strategy based on nanomaterials for leishmaniasis, a neglected tropical disease, which meets the current "One Health" requirements. The technical combination is novel: for the first time, sialic acid (SA, a host cell mimic) and colchicine (COL, a cytotoxic drug) are jointly used in a nanocarrier, enhancing selectivity through a dual mechanism. However, there are some shortcomings.

1.1.Lack of synergistic effect evaluation: No comparison was made between the activities of MSN-SA-COL and free SA+COL, making it impossible to confirm the synergistic advantages of the nanocarrier. No relevant statistical analysis or graphs were provided for this comparison.

We agree with the referee that more tests could help to stablish better comparisons between free drugs and the loaded ones in the proposed material. Even more tests with MSN+SA+COL or MSN-COL could give us interesting clues to confirm even more our hypothesis, or different percentage loadings of drugs in the silica, in the search of the most effective nanomaterial, but all of them must be performed in future experiments. This article is the first approach to this new nanocarrier and its effectiveness, and with our current resources we cannot perform all of the desired experiments. But we believe that these data support the conclusions of the paper, due to isolated SA has no significant antileishmanial activity.

For transparency, we add a paragraph in the discussion section clarifying that these are the preliminary studies and further combinations of the nanomaterial should and will be tested in the near future.

2.Incomplete testing scope:Only promastigotes (in vitro) were tested, lacking validation of intracellular amastigotes which better represent the clinical infection stage.

Axenic amastigotes were also tested, in table 3.

3.Unit inconsistency:IC50 values in the table are presented in both mg/mL and μM, compromising data comparability (e.g., when comparing Glucantime).

It is explained in the text below, explaining that there is not possible to talk about molarity in the nanomaterial due to its variety of composition, but if we refer only to one compound (colchicine or sialic acid) we can talk about µM and compare. The selectivity index is based in that reference in µM in the text, as well as marked with an * in the table to mark that there is a mathematical correction to compare active compound amount properly.

4.Incomplete technical details:(1)Reproducibility of synthesis: The consistency of particle size between nanoparticle batches is not described (TEM shows a wide distribution of 80-230 nm).(2)Low loading efficiency: COL loading is only 5%, without discussing optimization potential (e.g., pH adjustment or surface modification).

1) Despite the size of nanoparticles can be controlled depending on the protocol synthesis, it is not possible to obtain a completely homogeneous sample in terms of size. In turn, a distribution of sizes is provided in Figure 1 with a mean value around 100 nm.

2) The low COL loading was intentioned, not a technical limitation, just to see if a minimal amount of active compound in the whole nanomaterial have a significative leishmanicidal effect. As we say, we can perform further studies with many different COL loading, searching for the most active and worthy cost-effective relation.

5.Subheading 2.1.1, 2.1.2: Inconsistent Writing Style.

Style corrected

6.Write the references in accordance with the format of this journal.

7.Strengthen the discussion section.

8.Present the content in accordance with the format specifications of this journal.

We revised the format, added years, article number and DOIs missing, and improve the discussion section.

Reviewer 2 Report

Comments and Suggestions for Authors
  • The idea is intriguing; however, the main limiting point is that the study was performed only in an in vitro model without confirming the biological effect in an in vivo system. Furthermore, the study used only Leishmania braziliensis and ignored the other main species (e.g., L. donovaniL. major) that have economic importance.
  • No mechanistic studies were performed to support the drug effect.
  • The study reported low loading capacity. Are the authors able to discuss how they were able to overcome this point?
  • Many references are old.
    recommend some references for this manuscript, hope you can enrich your work with new studies like:
    https://doi.org/10.1016/j.bioorg.2024.108083,
    https://doi.org/10.3390/md23010016,
    https://doi.org/10.1016/j.actatropica.2024.107291

Author Response

  • The idea is intriguing; however, the main limiting point is that the study was performed only in an in vitro model without confirming the biological effect in an in vivo system. Furthermore, the study used only Leishmania braziliensis and ignored the other main species (e.g., L. donovaniL. major) that have economic importance.

We appreciate the reviewer suggestion and understand the point. However, as the title indicates, this work represents “A Novel Approach” based on nanomaterial functionalized which has been chemically characterized. And the bioactivity results are preliminary but show promising effects in achieving “Selective Treatments” which are part of the future perspectives.

This work is focused on L. braziliesis because is the principal causal agent of diffuse leishmaniasis which can produced hundreds or thousands of polymorphic skin lesions in various body regions, and frequent involvement of the nasal mucosa. So, the pathology is highly stigmatizing and even potentially fatal. Also, this affection is emerging out of Brazil and South America, reaching Europe and in the Middle East (doi: 10.1590/abd1806-4841.20198775). In any case, any clinical form of leishmaniasis is important since this clinical syndrome, unfortunately, is a neglected disease, so its importance is not assessed economically (https://www.who.int/health-topics/neglected-tropical-diseases#tab=tab_1).

Finally, one of the authors institution and funding provider is a L. braziliensis endemic area (Brazil), which also motivates us to select this specific strain.

  • No mechanistic studies were performed to support the drug effect.

The data compiled in the manuscript support our hypothesis, but we agree that mechanistic studies will enrich the article. However, this is the first approach to this novel nanomaterial and the purpose of the paper is to establish the bases to develop further experiments, focusing on concrete aspects of the effectiveness of the nanosystem. We also are limited by our current resources to perform more specific experiments, but we hope to reach fundings after this first step for a promising future against neglected diseases with the proposed kind of materials.

  • The study reported low loading capacity. Are the authors able to discuss how they were able to overcome this point?

The low COL loading was intentioned, not a technical limitation, just to see if a minimal amount of active compound in the whole nanomaterial have a significative leishmanicidal effect. As we say in expanded new discussion section, we can perform further studies with many different COL loading, searching for the most active and worthy cost-effective relation.

  • Many references are old.
    recommend some references for this manuscript, hope you can enrich your work with new studies like:
    https://doi.org/10.1016/j.bioorg.2024.108083,
    https://doi.org/10.3390/md23010016,
    https://doi.org/10.1016/j.actatropica.2024.107291

We thank your suggestions, and now are included in the text.

Reviewer 3 Report

Comments and Suggestions for Authors

Adan and coworkers built a vector that might kill Leishmania protozoa. After careful review I concluded that the study only characterized the vector in terms of physical and chemical properties and lacked an actual evaluation of the effectiveness of pathogen removal. I recommend rejection of the manuscript because the study is incomplete.

Author Response

Adan and coworkers built a vector that might kill Leishmania protozoa. After careful review I concluded that the study only characterized the vector in terms of physical and chemical properties and lacked an actual evaluation of the effectiveness of pathogen removal. I recommend rejection of the manuscript because the study is incomplete.

We thank the reviewer for taking time for reading our work, and we are sorry that he didn’t like it. Due biological test for extra and intracellular are performed and presented in the text, and the manuscript also contains the full physical and chemical characterization of a novel and effective nanomaterial, which is also accepted for revision fulfilling with all the formal criteria of characterization and biological tests, and currently under evaluation for a national patent as we say in the text, we assume that the rejection criteria is only based in personal feelings of the reviewer. Against that, authors can’t do much more than thank again the reviewer for his time and hoping to give more constructive revisions in the future.

Round 2

Reviewer 1 Report

Comments and Suggestions for Authors

The author has made improvements to the questions raised.

Comments on the Quality of English Language

/

Author Response

Thank you for your help to improve the manuscript

Reviewer 2 Report

Comments and Suggestions for Authors

The authors followed the comments.

Author Response

(The authors gave the same response as above.)

Reviewer 3 Report

Comments and Suggestions for Authors

General Comment:
This manuscript proposes a novel nanomaterial (MSN-SA-COL) for targeted Leishmaniasis treatment. While the concept of receptor-mediated drug delivery is innovative, the study suffers from fundamental flaws in biological rationale, experimental design, and data interpretation. The claims of selectivity and efficacy are inadequately supported, methodological gaps undermine reliability, and critical controls are missing. The work does not meet the standards required for publication in a high-impact biomedical journal. Major revisions would be necessary to address the core issues, but the pervasive weaknesses suggest rejection is appropriate. While nanomedicine approaches for neglected diseases are promising, this work requires a complete redesign—beyond the scope of revision.

Specific comments

  1. The premise that SA acts as a key receptor for Leishmania attachment is unsupported by established parasitology. Leishmania spp. primarily bind to macrophages via complement receptors (CR3, CR1), fibronectin receptors, or mannose-fucose receptors—not SA. That means the entire targeting strategy (SA as "bait") lacks biological plausibility. Without robust evidence that L. braziliensis promastigotes bind SA, the proposed mechanism is speculative.
  2. MSN-SA-COL shows 75% weight loss, implying 25% organic loading. However, SA and COL decompose at different temperatures (SA: ~200°C; COL: ~280°C), yet TG curves lack distinct mass-loss steps. This suggests incomplete decomposition or impurities, not quantified loading.

  3. Nanoparticle concentrations equate to ~10–200 µg/mL of COL. Free COL ICâ‚…â‚€ against promastigotes is 19.68 µM. The nanosystem (ICâ‚…â‚€: 0.055 mg/mL) delivers sub-therapeutic COL doses yet claims enhanced efficacy—a contradiction.

  4. Selectivity Index calculations ignore nanoparticle-induced toxicity (e.g., silica-induced inflammation), inflating apparent selectivity.
  5. Lack of PK-PD experiment.
  6. No assessment of parasite-nanoparticle binding by microscopy or flow cytometry

Author Response

This manuscript proposes a novel nanomaterial (MSN-SA-COL) for targeted Leishmaniasis treatment. While the concept of receptor-mediated drug delivery is innovative, the study suffers from fundamental flaws in biological rationale, experimental design, and data interpretation. The claims of selectivity and efficacy are inadequately supported, methodological gaps undermine reliability, and critical controls are missing. The work does not meet the standards required for publication in a high-impact biomedical journal. Major revisions would be necessary to address the core issues, but the pervasive weaknesses suggest rejection is appropriate. While nanomedicine approaches for neglected diseases are promising, this work requires a complete redesign—beyond the scope of revision.

Specific comments

1. The premise that SA acts as a key receptor for Leishmania attachment is unsupported by established parasitology. Leishmania spp. primarily bind to macrophages via complement receptors (CR3, CR1), fibronectin receptors, or mannose-fucose receptors—not SA. That means the entire targeting strategy (SA as "bait") lacks biological plausibility. Without robust evidence that L. braziliensis promastigotes bind SA, the proposed mechanism is speculative.

Sialic acids play a crucial role in the leishmania infection (as documented in references like Front. Cell. Infect. Microbiol., 2021, https://doi.org/10.3389/fcimb.2021.671913 or PLOS https://doi.org/10.1371/journal.pntd.0004904). We rewrite the introduction sentences to avoid misunderstanding in the “bait” proposal mechanism, due of course SA is not one primarily bind to macrophages, but it is necessary for leishmania to capture and cover with it to be phagocytized by the infected cells.

2. MSN-SA-COL shows 75% weight loss, implying 25% organic loading. However, SA and COL decompose at different temperatures (SA: ~200°C; COL: ~280°C), yet TG curves lack distinct mass-loss steps. This suggests incomplete decomposition or impurities, not quantified loading.

Decomposition temperature of colchicine is 160, not 280 C (according with Sigma SDS) and sialic acid around 180 oC. You can notice that the organic molecules are loaded in silica pores, which can alter the real decomposition temperature. Due to these two reasons, it is not realistic to try to observe a significative difference in the curve shapes, and they are evaluated as final weight difference, at very superior temperatures than theoretic decomposition temperature, to avoid interpretation errors.

3. Nanoparticle concentrations equate to ~10–200 µg/mL of COL. Free COL ICâ‚…â‚€ against promastigotes is 19.68 µM. The nanosystem (ICâ‚…â‚€: 0.055 mg/mL) delivers sub-therapeutic COL doses yet claims enhanced efficacy—a contradiction.

Nanoparticle concentrations of COL equate to 5% of the whole weight, so in a 0.055 mg/mL IC50, the COL contribution is 0.0025 mg/mL, which is the same to say 0.00625 µmol/mL. If we compare 6.25µM with 19.65 µM, the nanoparticle provides a better IC50 value. So yes, we can maintain that the MSN-SA-COL loaded colchicine is more effective than the free one.

4. Selectivity Index calculations ignore nanoparticle-induced toxicity (e.g., silica-induced inflammation), inflating apparent selectivity.

The toxicity in mammalian cells is taken into account as indicated in the table footers: bSelectivity index = IC50 against Vero cells / IC50 parasite (promastigote forms). Therefore, the toxicity of each compound on mammalian cells is taken into account.

5. Lack of PK-PD experiment.

Yes, there is no PK-PD experiment as well there is no in vivo test, for example. This article is the first proposal of a novel approach to leishmaniasis fight with a non-studied nanosystem, so authors designed experiments to prove the hypothesis. A lot of more specific experiments and lines of action can evolve from this first publication in the future. It is not realistic to expect that all possible biological tests can be carried out in a first article that proposes a novel route of action.

6. No assessment of parasite-nanoparticle binding by microscopy or flow cytometry.

The difference of size between parasite and nanoparticles make the microscopy images not valuable to observe the binding between MSN-SA-COL and promastigote or amastigote forms. Of course, more complex experiments can be performed in the future to study the exact nanoparticle-parasite mechanisms.

Here you can find attached and in supplementary material section some images that we measure, but authors consider them not valuable for the hypothesis confirmation, and they are not included in the main text.

Figure --. Leishmania braziliensis promastigote-nanoparticle binding by light microscopy. (A) Control of L. braziliensis promastigote in vitro culture, (B) and (C) promastigote of L. braziliensis promastigote treated with MSN-AS at 0,1 mg/ml, (D) promastigote of L. braziliensis promastigote treated with MSN-AS-COL at 0,1 mg/ml, and (E) promastigote of L. braziliensis promastigote treated with MSN-AS-COL at 0,5 mg/ml. Bar: 10 µm.

Round 3

Reviewer 3 Report

Comments and Suggestions for Authors

The authors have revised the MS according to my comments and addressed my concern. I am pleased to recommend its acceptance.